# *Cupressus sempervirens* Essential Oil, Nanoemulsion, and Major Terpenes as Sustainable Green Pesticides against the Rice Weevil

**Abdulrhman A. Almadiy [1], Gomah E. Nenaah [1,2,*] , Bader Z. Albogami [1], Dalia M. Shawer [3] and Saeed Alasmari [1]**

[1] Biology Department, College of Science and Arts, Najran University, Najran 1988, Saudi Arabia; aaalmady@nu.edu.sa (A.A.A.); bzalmarzoky@nu.edu.sa (B.Z.A.); smalasmari@nu.edu.sa (S.A.)
[2] Zoology Department, Faculty of Science, Kafrelsheikh University, Kafrelsheikh 33516, Egypt
[3] Economic Entomology Department, Faculty of Agriculture, Kafrelsheikh University, Kafrelsheikh 33516, Egypt
* Correspondence: dr_nenaah1972@yahoo.com

**Abstract:** In order to find effective, biorational, and eco-friendly pest control tools, *Cupressus sempervirens* var. *horizontalis* essential oil (EO) was produced using hydrodistillation, before being analyzed with gas chromatography, specifically, using flame ionization detection. The monoterpene components *α*-pinene (46.3%), *δ*-3-carene (22.7%), and *α*-cedrol, a sesquiterpene hydrocarbon, (5.8%), were the main fractions. An oil-in-water nanoemulsion was obtained following a green protocol. The EO, its nanoemulsion, and its terpenes each exhibited both insecticidal and insect repellent activities against the rice weevil, *Sitophilus oryzae*. In a contact bioassay, the nanoemulsion induced a 100% adult mortality rate in a concentration of 10.0 μL/cm$^2$ after 4 days of treatment, whereas 40 μL/cm$^2$ of EO and *α*-cedrol was required to kill 100% of weevils. Using fumigation, nanoemulsion and EO at 10 μL/L air caused a 100% adult mortality rate after 4 days of treatment. The LC$_{50}$ values of botanicals ranged between 5.8 and 53.4 μL/cm$^2$ for contact, and between 4.1 and 19.6 μL/L for fumigation. The phytochemicals strongly repelled the weevil at concentrations between 0.11 and 0.88 μL/cm$^2$, as well as considerably inhibiting AChE bioactivity. They were found to be safe for earthworms (*Eisenia fetida*) at 200 mg/kg, which also caused no significant alteration in wheat grain viability. This study provides evidence for the potential of using the EO of *C. sempervirens* and its nanoemulsion as natural, eco-friendly grain protectants against *S. oryzae*.

**Keywords:** cypress oil; nanoemulsion; terpenes; *Sitophilus oryzae*; bioactivity; biosafety

## 1. Introduction

The world's population is expected to grow to more than 10 billion by 2050, which will boost the global food demand; particularly, for food crops and cereal products. Therefore, agricultural production should be doubled if we are to secure adequate food sources for this huge number of people [1]. There is no doubt that many countries of the world will face problems in this regard, especially those developing and underdeveloped countries with poverty and inadequate technologies for modern agriculture. Furthermore, harmful insects and other pathogens cause the loss of more than 30% of the world's food production, both in the field and during storage as well [2,3]. Coleopteran arthropod insect pests are the major causal agents of grain loss during storage, worldwide. Grain weevils of the genus *Sitophilus* are among the world's most damaging and widespread pests for stored grains. Infestations of grains by these weevils cause significant grain losses and promote the growth of molds, including harmful toxigenic species, by increasing temperature and moisture levels [3].

The rice weevil, *Sitophilus oryzae* L. (Coleoptera: Curculionidae), is a serious primary internal feeder pest, able to infest intact grains, causing both quantitative and qualitative

damage in the grains and altering seed viability [4]. Chemical insecticides have widely been applied to prevent insect pests from attacking a variety of crop plants, both in the field and in stores. The overuse of chemical insecticides negatively affects not only the health of consumers and farmers, but also nontarget beneficial organisms and the quality of foodstuffs, in addition to bringing environmental problems as well [5,6]. Populations of *S. oryzae* have also developed resistance against many insecticides and fumigants, such as pirimiphos-methyl and phosphine [7]. These problems make the intensive use of chemical pesticides a point of renewed public debate, especially in light of increasing public awareness of the risks of insecticides. This popular awareness of the adverse effects of conventional pesticides has promoted researchers to seek novel agrochemical pesticides that can meet the increasing grower, consumer, environmental, and regulatory requirements [8]. These new pesticides should be environmentally safe, with novel action mechanisms and low negative impacts on the ecosystem. Pest control using phytochemicals, especially plant essential oils (PEOs), seems a promising alternative strategy for decreasing the intensive reliance on conventional insecticides [6,9]. Because of their wide spectrum bioactivity as pest control tools against several pest insects (including those of stored grains), PEOs are increasingly being considered a credible eco-friendly natural alternative to conventional insecticides [3,6,9–14]. This is especially significant for the protection of stored products, whose confinement promotes the action of molecules which are naturally highly volatile, in order to avoid the toxic residue from chemical insecticides contaminating food products. In this context, we can observe major shortcomings related to the procedures adopted for the extraction, formulation, application, and performance of plant products (including EOs) in pest control protocols. For strengthening and improving the inclusion of plant-based products in pest control programs, nanotechnology has emerged as a promising field across multidisciplinary research, opening up many application opportunities in such various fields as medicine, electronics, drug delivery, and agriculture [15,16]. In agriculture, the potential benefits of nanotechnology in pest control programs include the fabrication of novel plant-based nanomaterial formulations with enhanced insecticidal activities, which could diminish the need for repeated application of chemical pesticides [6,15–18]. Because of their high reactivity and solubility, in addition to their novel physical and chemical characteristics, nanopesticide materials show enhanced bioactivities as pest control tools relative to their bulk counterparts [6,15,17].

The genus Cupressus (Cupressaceae) comprises about 12 species, spread across North America, Mexico, the Mediterranean basin, southern Europe, and subtropical west Asian countries, including the Kingdom of Saudi Arabia [19]. Members of Cupressaceae are common essential-oil-bearing plants, of which an important species is *Cupressus sempervirens* L. var. *horizontalis* (The Mediterranean cypress). It is an aromatic evergreen tree, traditionally used as an expectorant in anticough and antibronchitis medications; for stomach pain; as an antidiabetic medicine; as an antiseptic; for antiulcer and anti-inflammatory purposes; and for treatment of toothaches, flu, coughs, and laryngitis [20]. This plant species has been screened for different bioactivities, including antimicrobial, antiviral, antihelmenthic, antiseptic, cytotoxic, antioxidant, anti-inflammatory, antirheumatic, antihyperlipidemic, anticancer, antispasmodic, antidiuretic, and hepatoprotective activities [20,21]. Insecticidal bioactivities of the cypress tree have also been recorded [9,22,23]. However, there have been no thorough investigations into the insecticidal bioactivity of the EO, nanoemulsion, or bioactive terpenes of *C. sempervirens* against stored-grain insects. In this study, we aimed to investigate not only the composition, but also the contact, fumigant, and repellence bioactivities of the oil, nanoemulsion, and bioactive terpenes of cypress trees growing in Saudi Arabia against *S. oryzae*. The effect of EO materials on acetylcholinesterase (AChE) bioactivity—being a common target enzyme for insect control agents—was investigated. The impact of EO products on earthworms (*E. fetida*), as well as their phytotoxic activity on wheat plant in terms of the basic growth parameters (%germination, root, and shoot growth) were evaluated.

## 2. Materials and Methods

### 2.1. Chemicals

Analytical grade monoterpenes, sesquiterpenes, and oxygenated monoterpenes, (Sigma-Aldrich Co. Ltd., St Louis, MO, USA; label purity 99.0–99.8%) were used for comparisons. To calculate retention indices, the series of hydrocarbons ($C_5$–$C_{40}$) known as triacontane (Supelco, Bellefonte, CA, USA) was used. Dimethyl sulfoxide (DMSO) and its solvents, all of an analytical grade (Carlo Erba Milan, Milan, Italy), were used in experiments.

### 2.2. Test Insect

A laboratory strain of the rice weevil, the *S. oryzae* were reared in a pesticide-free environment in our laboratory for more than twenty generations. Weevils were kept in 2 L glass bottles, each containing 200–250 adults and 250 g sterilized wheat grains (moisture content $14 \pm 2\%$). The jars were covered using muslin cloth that was held in place with a rubber band, and then kept in laboratory conditions of $30 \pm 2\,°C$ and 68.5% relative humidity, in complete darkness, until the emergence of the adult specimens.

### 2.3. EO Extraction

On September 2020, the whole aerial parts of the cypress trees were collected from random gardens in Abha, Kingdom of Saudi Arabia; latitude $18°\,19'\,45.7824''$ (N), longitude $42°\,45'\,33.7140''$ (E), and 718 m altitude. Specimens were identified by the botanists of Najran University, Saudi Arabia. A plant specimen (No. Cs 02) was laboratory-deposited, for reference. Plant leaves were air dried in the shade, powdered mechanically using a high-speed blender, before being subjected to hydrodistillation using a Clevenger apparatus to obtain the EO. In each run, 150 g powders were hydrodistilled for 3 h with 250 mL distilled water, and in three replicates. The oil/water mixture was extracted with hexane, which had been washed with anhydrous sodium sulfate before the oil had dried, before then being concentrated under reduced pressure. For bioassays, the oil yield (% *wt./wt.*) was calculated on a dry weight basis and stored at $4\,°C$, where triplicates were considered in calculating the oil yield.

### 2.4. Analysis of EO and Identification of Constituents

Cypress oil was analyzed using an Agilent 6890 N gas chromatograph (Agilent Technologies, Palo Alto, CA, USA), coupled with a flame ionization detection (FID) and an HP-5 capillary column (30 m 0.32 mm; thickness 0.25 m). The following conditions were met by the GC–FID: one liter of EO; split mode; 50:1 split ratio; and an injector temperature of $250\,°C$. Oven temperature was initially set at $40\,°C$ for 3 min, then increased to $80\,°C$ at a rate of $5\,°C/min$, held at that temperature for 3 min, then increased to $250\,°C$ at $10\,°C/min$, which was held for 10 min. The injector and detector were set to $250\,°C$, and the carrier gas was helium, at a 1.0 mL/min flow rate. The gas chromatograph was then connected to the silica gel capillary column (HP-5 MS). At a split ratio of 1:100, 0.1 μL of EO was injected onto the column; the carrier gas was helium (1.0 mL/min flow rate). Operation of the mass detector was set at 70 eV ionization voltage. The mass range was taken at 45~550 AMU. Temperatures of the ion source, transfer-line, and the quadrupole were 230, 250, and $150\,°C$, respectively. Temperature of the oven was programmed as described for the GC. Retention indices of terpenes were calculated depending on *n*-alkanes ($C_5$–$C_{40}$) co-injected into the column, in accordance with Van Den Dool and Kratz's equation. Identification of EO profile was accomplished by comparing terpene retention indices and mass spectra to those recorded by Adams [24] and the data stored in the database NIST Standard Reference Database Number 69, [25]. The oil terpenes were quantified as percentages by integrating their peak areas, calibrating, and comparing them to internal standards without the use of a response factor correction. The remaining terpenes were likewise quantified as percentages, calculated by integrating their peak areas, calibrating, and then comparing to standards without using a response factor correction.

### 2.5. Isolation of Main Terpenes

Ten milliliters of cypress oil was fractionated on a silica gel capillary column (Kieslgel 60, 230–400 mesh, Merck). Trials were conducted in order to determine the best eluent. To accomplish this, several solutions of both n-hexane: ethyl acetate and toluene: ethyl acetate were prepared, which were then tested using Thin-layer Chromatographic plates (TLC), with toluene: ethyl acetate (90:10, then 93:7) being chosen as the best eluent [26]. According to TLC data, developed fractions were divided into 3 main fractions: Main fraction 1 (fractions 8–14, 274.3 mg) was fractionated on a silica gel column, affording 41 mg of α-pinene. The main fraction 2 (fractions 21–27, 193.1 mg) yielded 23 mg of δ-3-carene. By contrast, when fraction 3 (fractions 31–37, 82.5 mg) was developed it provided 9.2 mg of α-cedrol. Terpene fractions were visualized using an UV lamp (254 and 365 nm), and the $R_f$ values of terpenes were calculated and then compared against standards. The structures of the terpene fractions were elucidated using spectroscopic instruments.

### 2.6. Nanoemulsion Formulation and Characterization

An oil/water nanoemulsion was made following a low-energy emulsification protocol at a constant temperature, with the following proportions: deionized $H_2O$ (90%), EO 5% (*wt./wt.*), and Tween 80 (5%) as a nonionic surfactant [17]. The EO/emulsifier mixture was stirred at 800 rpm for 30 min in a water bath at $35 \pm 5$ °C. After reaching an oily phase, deionized $H_2O$ was gradually added (2.5 mL/min). After stirring for 45 min at 800 rpm, the temperature was gradually reduced to room temperature, and the nanoemulsion was formed and then preserved in dark screw-capped vials at $24 \pm 2$ °C. At 0, 1, 10, 20, 30, and 45 days following its preparation, the nanoemulsion was examined for thermodynamic changes. Nanoemulsion was characterized using a polydispersity index (PDI), mean droplet size, and thermodynamic stability measurements (centrifugation, heating, cooling, freezing cycles, and viscosity) [17]. The oil emulsion was first centrifuged (5000 rpm at 25 °C for 25 min), and checked for phase separation, turbidity, and cracking, if any. Heating/cooling tests were performed on the stable formulations for 6 cycles (4–40 °C, each cycle lasting 48 h). Emulsions that demonstrated stability were subjected to a freeze–thaw stress test by alternately storing them at 2 different temperatures (20 °C and 20 °C, 24 h each). Nanoemulsions that demonstrated stability were stored in dark, tightly closed vials at room temperature for one month to observe any creaming, phase separation, or flocculation. The emulsion's pH was measured at $25 \pm 0.2$ °C, and its viscosity (μ) was elucidated at 200 rpm. Samples were left to stand for about 2 min to reach an equilibrium; thus, readings and experiments were each taken and performed in triplicate. The Z-average diameter of droplets and PDI were measured using a nanoparticle analyzer apparatus (Zetasizer, Nano ZS, Malvern Instruments, Worcestershire, UK) that operates on a dynamic light-scattering basis. Prior to measurements, test formulations were diluted to 10% with deionized $H_2O$ to avoid multiple scattering. The droplets' size and their PDI were calculated using DLS data. The measurements were taken at a scattering angle of 90°, and trials were repeated three times. A Scanning Electron Microscope, or SEM (JEOL, JFC-1600, Tokyo, Japan), was used to determine the morphology of droplets; for this, 15 μL of emulsion that had been dissolved in deionized $H_2O$ was placed onto a carbon-coated copper grid that had been stained with 2% phosphotungstic acid (pH of 6.8). The test samples were dried at 26 °C before being imaged at 80 kV.

### 2.7. Contact Insecticidal Activity

The contact bioactivity of the EO, nanoemulsion, and terpenes from cypress trees against *S. oryzae* adults were determined using the dipping filter paper technique [6]. Test concentrations of 0.398, 0.795, 1.59 and 3.18 mL of EO, nanoemulsion, δ-3-carene, α-cedrol, and α-pinene were dissolved in 5 mL acetone to obtain the test solutions. Each test concentration was uniformly dropped onto a filter paper (Whatman No. 1, 9 cm d, 63.6 cm$^2$) to achieve serial test concentrations of 5.0, 10.0, 20.0, and 40.0 μL/cm$^2$, respectively. Acetone was evaporated from the treated papers; then a treated paper was placed into the bottom

of a Petri dish (9 cm d) and twenty adult weevils (of mixed sexes, and 15–20 days old) were introduced. Control groups (adults exposed to acetone-treated filter papers) were included. Treatment and control dishes were kept in the dark at 30 ± 2 °C and 68 ± 5% r.h. After 24 h, insects were placed into clean Petri dishes, enriched with wheat grains, and kept in rearing conditions. Experiments were performed six times alongside control, and mortality was recorded 1, 2, 4 and 7 days after the treatment, whereafter the end-point mortality was reached, and the resulting contact toxicity was expressed in $\mu L/cm^2$.

### 2.8. Fumigation

Insecticidal bioactivity using fumigation was investigated as follows: a filter paper (7.0 cm diameter) was dipped in 25 μL of an appropriate concentration of each terpene dissolved in acetone, and control sets were made using acetone, only [6]. Botanicals were screened at 4 dose rates (2.5, 5.0, 10.0, and 20.0 μL/L air). After evaporating the acetone, each treated paper was attached to the undersurface of the screw cap of 250 mL volume glass bottles, which served as fumigant chambers. Twenty weevils were placed into each bottle as adults (15–20 days old), and the bottle was covered with a tape-fixed fine gauze. Experiments were undertaken in six replicates, alongside control groups. After 24 h, the weevils were transferred back to food-enriched clean vials and kept in the rearing conditions described before; whereupon mortality was measured after 1, 2, 4 and 7 days had elapsed from treatment, and the resulting fumigant bioactivity was expressed in μL/L air.

### 2.9. Repellence Bioactivity

The repellence bioactivity of *C. sempervirens* oil materials against adult weevils was studied by adopting a chosen (area preference) bioassay [22]. A piece of filter paper (Whatman No. 1, 9 cm diameter) was divided into two halves. Test solutions of oil materials were prepared, with 3.5, 7.0, 14.0, and 28 μL of each material dissolved in 0.5 mL n-hexane. Each test concentration was uniformly dropped onto a half filter paper disc, which served as a test area, to obtain bioassay concentrations of 0.11, 0.22, 0.44, and 0.88 $\mu L/cm^2$. The second half was treated only with n-hexane, representing a control. The treated and untreated paper discs were then air dried for 5 min, and thereafter attached to their corresponding opposite surface with adhesive tape, and put in the bottom of a Petri dish (9 cm). Twenty (15–20 days old) unsexed adult weevils of *S. oryzae* were released at the center of each disc, then the lid was covered using a parafilm. Five replicates (100 adults) were considered for each concentration, and the experiments were achieved in rearing conditions. The number of weevils that were observed across both the treated and control halves were counted after 2, 6, 12, and 24 h. The Repellency percentage (RP) was calculated using the following formula: RP = (C − T)/(C + T) × 100, where C is the No. of weevils on untreated zone, and T is the No. of weevils on control zone.

### 2.10. AChE Inhibition and Estimation of $IC_{50}$

Anticholinesterase (AChE) enzymatic activity was measured in accordance with Ellman et al. [27]. One gram's worth of the adult weevils were homogenized in 20 mL of an ice-cold phosphate buffer (50 mM and pH 7.4). Acetylthiocholine iodide (25 μL of 15 mM) was dropped as a substrate. The inhibition in AChE activity was measured calorimetrically using a supernatant as an enzyme source [17]. Botanicals were formulated initially in acetone, then in Triton-X 100 (0.01%), and were then tested at 2.5~100 mM. Test and control solutions were corrected using blanks for the nonenzymatic hydrolysis. Trials were performed in triplicate. Absorbance of the solution reflecting AChE specific activity (ΔOD/mg protein/min) was monitored at a wavelength of 412 nm.

### 2.11. Phytotoxicity

The phytotoxic impact of the oil terpene components (as indicated by basic growth parameters (%germination, root, and shoot growth)) was evaluated on wheat plants (*Triticum aestivum* L.). Wheat seeds were sterilized using a solution of sodium hypochlorite (15%) for

about 40 s, followed by rinsing in sterile deionized water. The grains were placed in clean 9 cm diameter Petri plates, each containing five layers of Whatman filter paper, onto which 1 mL of each botanical (at concentrations of 50, 100, and 150 µL/mL) was dropped. 2 mL of methanol was sprayed on the control. After the evaporation of methanol, ten healthy grains (~0.3–0.36 g) were deposited in each dish. Dishes were maintained at 20 ± 2 °C; 65 ± 5% R.H., with a natural photoperiod (optimized environmental conditions for wheat germination). Additionally, 10 mL of water was given daily. Each concentration had five replicates, as well as a control. Germination and the seedling growth were noticed after 10 days of planting. The length of shoot and number of leaves were counted 2 weeks later, and seed germination was indicated by the emergence of radicles.

### 2.12. Toxicity on Earthworm

The acute toxicity of the botanicals against *E. fetida* earthworms was investigated, according to the guidelines of OECD (Organization for Economic Co-Operation and Development [28]. The animals were reared on artificial diet, as detailed by Pavela [29]. Terpenes were admixed with the soil at concentrations equaling 50, 100, and 200 mg kg$^{-1}$. The positive control was $\alpha$-cypermethrin at 10 and 20 mg kg$^{-1}$ soil alongside, with deionized water as a negative one. In 1 L glass pots containing either treated or untreated (control) soil, the earthworms were confined as ten adults, and triplicates were made for each run. Pots were covered with a fine gauze, then incubated at 22 ± 2 °C, 75 ± 5% R.H., and 16:8 h light/dark photoperiod. Mortality was recorded after five and ten days of treatment.

### 2.13. Statistical Analysis

Data of mortality were adjusted for control mortality and corrected using Abbott's formula [30] when mortality in control exceeded (5%), and data were expressed as % means (±S.E.). A one-way analysis of variance (ANOVA) at the probability level (=0.05) was adopted on transformed data to compare significance differences between means in both the treatment samples and the controls, followed by individual pairwise comparisons, adopting Tukey's HSD test. Dose–response mortality was analyzed using Finney's Probit analysis to estimate the $LC_{50}$ and $LC_{95}$ and their limits across 48 exposure periods [31]. Probit analysis was adopted to calculate the concentrations that inhibited AChE bioactivity by 50% ($IC_{50}$). The Statistical Package for Social Sciences was used for data analysis (version 23.0; SPSS, Chicago, IL, USA).

## 3. Results

### 3.1. Composition of EO

A pale yellowish EO with a strong odor (yield 0.74% *w/w*) was obtained from *C. sempervirens* var. *horizontalis* using hydrodistillation. A total of 62 terpenes amounting 99.7% (*wt./wt.*) were identified in the oil (Table 1 and Figure 1a), and then listed according to their retention indices. The main oil terpenes were (1*S*,5*S*)-2,6,6-trimethylbicyclo [3.1.1] hept-2-ene ($\alpha$-pinene, 46.3%), 3,7,7-trimethylbicyclo [4.1.0] hept-3-ene ($\delta$-3-carene, 22.7%), and (1*S*,2*R*,5*S*,7*R*,8*R*)-2,6,6,8-tetramethyltricyclo [5.3.1.0] undecan-8-ol ($\alpha$-cedrol, 5.8%) (Figure 1b). The structure of the terpenes was confirmed using physical and spectroscopic methods, which corroborates with published data:

*α-pinene*

Colorless, $C_{10}H_{16}$. $^{1}$H NMR (CDCl$_3$, 300 MHz): $\delta$ 1.92 (m, 1H), $\delta$ 1.4 (s, 3H), $\delta$ 1.6 (s, 3H), $\delta$ 1.8 (s, 3H), $\delta$ 1.9 (m, 2H), $\delta$ 2.3 (m, 1H), $\delta$ 2.4 (m, 1H), $\delta$ 4.2 (s, 1H), $\delta$ 5.6 (t, 1H), 20.67, 22.35, 22.47, 26.36, 32.62, 36.08, 68.03, 94.15, 123.87, 134.38; $^{13}$C NMR (125 MHz, CHCl$_3$): $\delta$ 46.99 (C-1), 144.6 (C-2), 116.0 (C-3), $\delta$ 31.3 (C-4), 40.69 (C-5), 37.97 (C-6),31.5 (C-7), $\delta$ 26.3 (C-8), 20.8 (C-9), 23.01 (C-10) [32,33].

*δ-3-carene*

Colorless, $C_{10}H_{16}$. $^{1}$H-NMR (600 MHz, CDCl$_3$) $\delta$ ppm: 5.23 (2H, t, H-2), 2.62 (2H, t, H-6), 1.60 (4H, m, H-3, 5), 1.022 (1H, m, H-4), 0.761 (6H, s, 9, 10-CH$_3$), 0.49 (3H, s, 7-CH$_3$);

$^{13}$C-NMR (125 MHz, CDCl$_3$) $\delta$ ppm: 131.30 (C-6), 119.56 (C-1), 28.42 (C-7), 24.93 (C-5), 23.63 (C-4), 20.89 (C-2), 18.71 (CH$_3$-8), 16.90 (C-3),16.78 (CH$_3$-9), 13.20 (CH$_3$-l0) [34,35].

*α-cedrol*

Colorless, C$_{15}$H$_{26}$O. $\nu_{max}$/cm$^{-1}$ 3380, 1460, 1030, and 1000; $\delta_H$(300 MH$_z$) 0.74 (3H, s, Me), 0.78 (3H, d, *J* 7.1, Me), 0.85 (3H, s, Me), 0.88 (3H, s, Me), 0.85–1.61 (10H, m), 1.87–1.98 (1H, m), 2.1–2.2 (1H, m), and 3.94 (1H, ddd, *J* 2.2, 5.6 and 9.7, CHOH); *m/z* 222 (M1, 44%), 206 (15), 178 (100), and 123 (40) (Found: M1, 222.1992. C$_{15}$H$_{26}$O requires *M*, 222.1985) [36–38].

**Table 1.** Chemical profile of *Cupressus sempervirens* essential oil.

| [a,b] Components | [c] RI exp. | [d] RI lit. | Concentration (%) |
|---|---|---|---|
| 2-Hexanal | 860 | 862 | 0.2 |
| Tricyclene | 918 | 916 | 0.1 |
| *a*-Thujene | 920 | 921 | 0.4 |
| *α*-Pinene | 928 | 930 | 46.3 |
| Camphene | 930 | 932 | 1.2 |
| *a*-Fenchene | 941 | 942 | 0.1 |
| Sabinene | 966 | 967 | 0.6 |
| *ß*-Pinene | 980 | 980 | 0.9 |
| *ß*-Myrcene | 988 | 988 | 0.1 |
| *α*-Phellandrene | 1006 | 1008 | 0.2 |
| *δ*-3-Carene | 1010 | 1010 | 22.7 |
| *α*-Terpinene | 1016 | 1018 | 1.3 |
| *p*-Cymene | 1021 | 1020 | 0.6 |
| Limonene | 1032 | 1029 | 1.6 |
| *β*-Phellandrene | 1034 | 1032 | 0.2 |
| *Z-ß*-Ocimene | 1038 | 1037 | 0.2 |
| *E-ß*-Ocimene | 1044 | 1044 | 0.1 |
| *γ*-Terpinene | 1054 | 1055 | 0.3 |
| *cis*-Sabinene hydrate | 1067 | 1066 | 0.4 |
| *p*-Cymenene | 1070 | 1072 | 0.9 |
| *α*-Terpinolene | 1085 | 1086 | 1.3 |
| Linalool | 1096 | 1095 | 0.4 |
| *trans*-Sabinene hydrate | 1099 | 1097 | 1.1 |
| Pinocarveol | 1138 | 1140 | 0.1 |
| Camphor | 1142 | 1144 | 0.2 |
| Pinocarvone | 1158 | 1162 | 1.6 |
| Borneol | 1162 | 1165 | 0.2 |
| Terpinen-4-ol | 1176 | 1174 | 1.1 |
| *p*-Cymen-8-ol | 1180 | 1181 | 0.4 |
| *trans*-Pinocarveol | 1182 | 1184 | 0.6 |
| *α*-Terpineol | 1188 | 1186 | 0.2 |
| Myrtenol | 1192 | 1195 | 0.2 |
| Pulegone | 1235 | 1233 | 0.3 |
| Carvacrol methyl ether | 1241 | 1241 | 0,2 |
| *cis*-Chrysanthenyl acetate | 1244 | 1242 | 0.1 |
| *cis*-Piperitone epoxide | 1247 | 1248 | 0.4 |
| *trans*-Piperitone epoxide | 1251 | 1252 | 0.7 |
| Carvone | 1254 | 1258 | 1.1 |
| Carvenone oxide | 1260 | 1260 | 0.2 |
| Bornyl acetate | 1285 | 1186 | 0.4 |
| Thymol | 1289 | 1288 | 0.3 |
| *trans*-Sabinyl acetate | 1290 | 1292 | 0.2 |
| Carvacrol | 1296 | 1298 | 0.3 |
| *α*-Cedrene | 1295 | 1294 | 0.1 |
| *α*-Copaene | 1372 | 1374 | 0.2 |
| *β*-Bourbonene | 1382 | 1384 | 0.1 |
| *α*-Gurjunene | 1408 | 1408 | 0.3 |
| *β*-Caryophyllene | 1414 | 1417 | 0.1 |

**Table 1.** *Cont.*

| a,b Components | c RI exp. | d RI lit. | Concentration (%) |
|---|---|---|---|
| β-Gurjunene | 1430 | 1432 | 0.2 |
| α-Humulene | 1450 | 1452 | 0.2 |
| Alloaromadendrene | 1472 | 1474 | 0.3 |
| Germacrene D | 1480 | 1478 | 0.3 |
| Bicyclogermacrene | 1496 | 1495 | 0.1 |
| β-Bisabolene | 1508 | 1510 | 0.2 |
| *cis*-Calamenene | 1544 | 1443 | 0.2 |
| Spathulenol | 1574 | 1576 | 0.6 |
| α-Cedrol | 1596 | 1591 | 5.8 |
| α-Acorenol | 1632 | 1630 | 0.3 |
| β-Acorenol | 1635 | 1637 | 0.2 |
| γ-Cadinol | 1648 | 1649 | 0.1 |
| Cadalene | 1674 | 1674 | 0.1 |
| Manool | 1990 | 1989 | 0.3 |
| Grouped compounds (%) | - | - | |
| Monoterpene hydrocarbons | - | - | 77.1% |
| Oxygenated monoterpenes | - | - | 12.9% |
| Sesquiterpene hydrocarbons | - | - | 9.7% |
| % peaks identified | - | - | 99.7 |
| Total yield % (mL/100 g) | - | - | 0.74 |

[a] Compounds are listed in the order of their elution from a HP-5MS column. [b] Identification methods: a, based on comparison of RT, RI, and MS with those of authentic compounds; b, based on comparison of mass spectrum with those reported in Wiley, Adams [24], and NIST 69 MS libraries [25]. [c] Linear retention index on the HP-5MS column, experimentally determined using homologous ($C_5$–$C_{40}$) *n*-alkane series. [d] Linear retention index based on Adams [24] or NIST 69 [25], and literature.

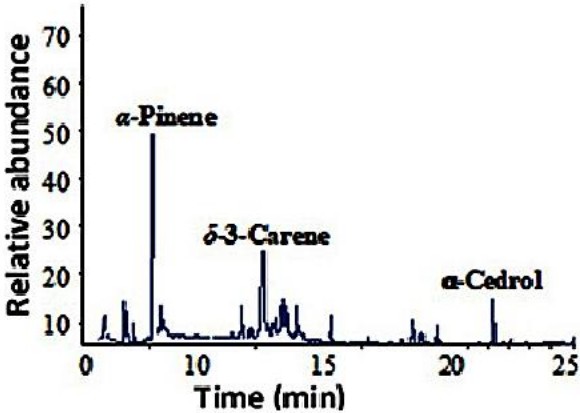

(a)

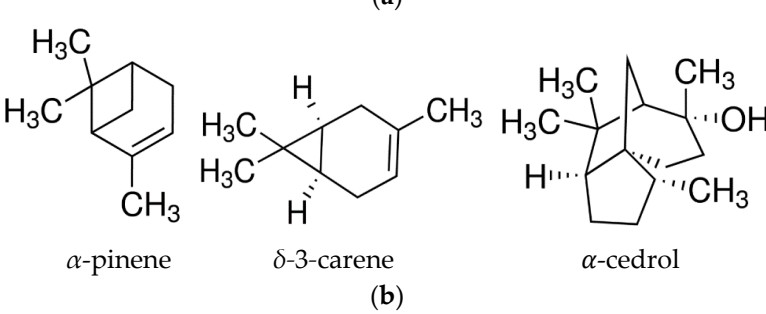

α-pinene        δ-3-carene        α-cedrol

(b)

**Figure 1.** (**a**) GC–FID chromatogram of cypress EO. Major terpenes are highlighted; (**b**) Major terpene components of cypress EO.

### 3.2. Nanoemulsion Characterization

The developed emulsion (droplet size 67.8 ± 3.1 nm) showed stability during extreme conditions of centrifugation, temperature, heating–cooling cycle (4–40 °C), and a freezing cycle at −4 °C. The optimum conditions of nanoemulsion preparation, droplet size, and their PDI, are listed in Table 2 and illustrated in Figure 2. The SEM revealed that a transparent nanoemulsion consisting of dispersed and spherical-shaped nanoparticles had been developed (Figure 3).

**Table 2.** Characterization of *Cupressus sempervirens* oil nanoemulsion.

| Storage Period (Days) | Viscosity (mPa·s) | pH | PDI | Size (nm ± S.E.) |
|---|---|---|---|---|
| 0 | 4.1 | 6.1 ± 0.04 [c] | 0.18 ± 0.03 [a] | 67.8 ± 3.1 [a] |
| 1 | 4.1 | 5.8 ± 0.06 [b] | 0.20 ± 0.05 [a] | 69.2 ± 3.6 [a] |
| 10 | 4.4 | 5.6 ± 0.04 [b] | 0.20 ± 0.02 [a] | 73.4 ± 4.2 [b] |
| 20 | 4.8 | 5.1 ± 0.08 [a] | 0.23 ± 0.03 [b] | 78.6 ± 5.7 [bc] |
| 30 | 5.1 | 4.9 ± 0.16 [a] | 0.25 ± 0.02 [bc] | 86.1 ± 6.3 [c] |
| 45 | 5.5 | 4.7 ± 0.14 [a] | 0.28 ± 0.02 [c] | 92.1 ± 6.1 [d] |

Each experiment is the mean of three replicates. Within a column, means followed by same letter(s) are not significantly different ($p \leq 0.05$).

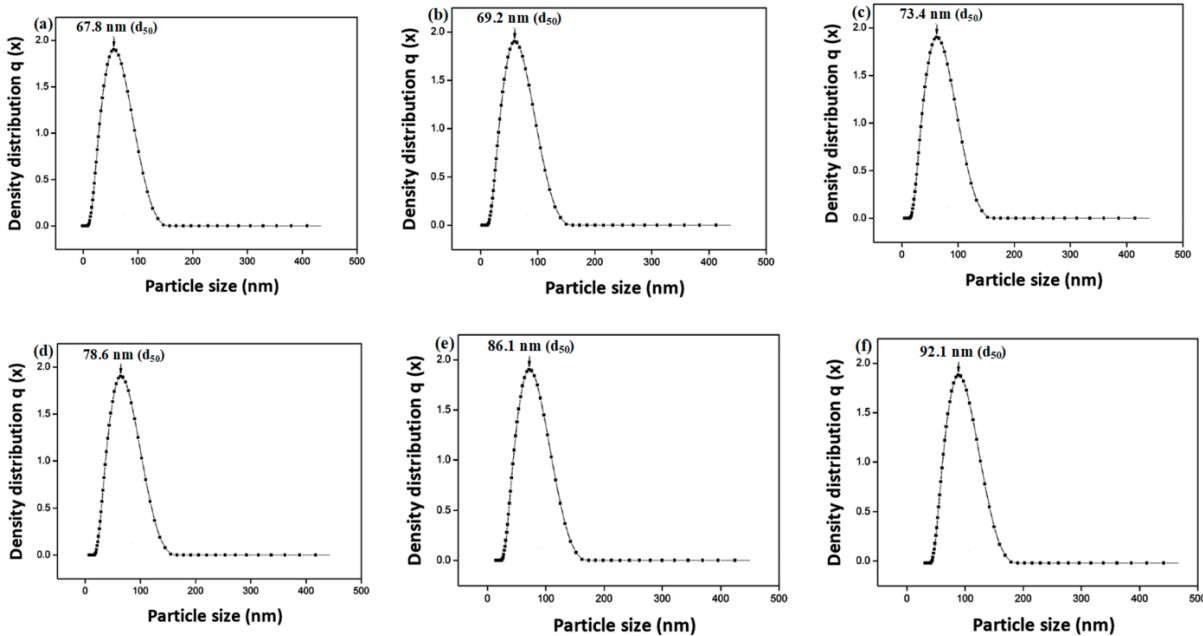

**Figure 2.** Particle size of nanoemulsion from cypress oil after: (**a**) 0 day, (**b**) 1 day, (**c**) 10 days, (**d**) 20 days, (**e**) 30 days, and (**f**) 45 days.

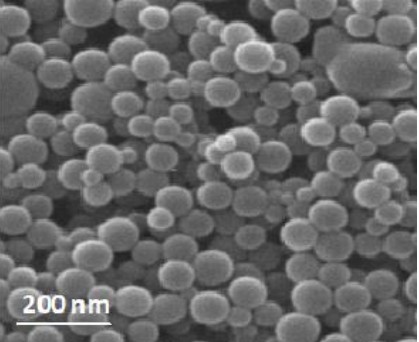

**Figure 3.** SEM of cypress oil nanoemulsion.

### 3.3. Contact Bioactivity

Contact bioactivity of EO materials was both dose- and time-dependent (Table 3). The oil/water nanoemulsion caused the strongest activity, with which 100% adult mortality of *S. oryzae* was reached at 10.0 µL/cm² at 4 days' following exposure. Under these conditions, the percentage mortality was 71.1, 60.3, 37.1, and 33.6% for EO, α-cedrol, δ-3-carene and α-pinene, respectively. After 7 days from exposing the weevils to 40 µL/cm² of botanicals, the mortality of the adults ranged between 77.5 and 100%.

**Table 3.** Contact insecticidal bioactivity of *S. oryzae* exposed to essential oil, nanoemulsion, and terpene fractions of *C. sempervirens*.

| Test Material | Concentration (µL/cm²) | Mortality (% Mean ± S.E.) after Exposure Period | | | |
|---|---|---|---|---|---|
| | | **Day 1** | **Day 2** | **Day 4** | **Day 7** |
| Crude oil | 5.0 | 18.1 ± 2.3 [ghi] | 25.3 ± 1.1 [fg] | 44.3 ± 2.6 [g] | 70.3 ± 2.1 [d] |
| | 10.0 | 27.6 ± 2.1 [f] | 46.6 ± 2.3 [e] | 71.1 ± 2.3 [d] | 84.1 ± 2.2 [c] |
| | 20.0 | 39.9 ± 2.1 [e] | 63.6 ± 3.1 [d] | 88.0 ± 3.2 [b] | 100.0 ± 0.0 [a] |
| | 40.0 | 71.3 ± 3.0 [b] | 93.8 ± 2.6 [a] | 100.0 ± 0.0 [a] | 100.0 ± 0.0 [a] |
| Nanoemulsion | 5.0 | 36.3 ± 3.1 [e] | 48.1 ± 2.1 [e] | 83.6 ± 3.1 b [c] | 92.4 ± 2.1 b [b] |
| | 10.0 | 54.4 ± 3.1 [d] | 70.9 ± 2.1 [c] | 100.0 ± 0.0 [a] | 100.0 ± 0.0 [a] |
| | 20.0 | 67.8 ± 2.6 [bc] | 81.3 ± 1.9 [b] | 100.0 ± 0.0 [a] | 100.0 ± 0.0 [a] |
| | 40.0 | 92.1 ± 3.3 [a] | 100.0 ± 0.0 [a] | 100.0 ± 0.0 [a] | 100.0 ± 0.0 [a] |
| α-Cedrol | 5.0 | 15.8 ± 1.1 [ijk] | 29.3 ± 2.1 [f] | 36.1 ± 2.1 [h] | 44.8 ± 2.1 [h] |
| | 10.0 | 25.5 ± 2.1 [fg] | 41.9 ± 2.3 [e] | 60.0 ± 2.1 [e] | 71.0 ± 2.1 [d] |
| | 20.0 | 36.0 ± 2.3 [e] | 60.8 ± 2.6 [d] | 70.9 ± 1.9 [d] | 83.3 ± 1.7 [c] |
| | 40.0 | 61.3 ± 3.1 [cd] | 83.1 ± 3.1 [b] | 100.0 ± 0.0 [a] | 100.0 ± 0.0 [a] |
| δ-3-Carene | 5.0 | 11.3 ± 2.0 [jk] | 16.6 ± 2.1 [h] | 22.6 ± 3.1 [i] | 29.3 ± 4.2 [j] |
| | 10.0 | 19.0 ± 3.0 [ghi] | 30.3 ± 1.9 [f] | 37.1 ± 2.1 [h] | 46.7 ± 2.4 [h] |
| | 20.0 | 27.3 ± 2.3 [f] | 41.9 ± 1.6 [e] | 54.3 ± 2.6 [ef] | 65.9 ± 2.0 [f] |
| | 40.0 | 38.2 ± 3.3 [e] | 60.4 ± 3.1 [d] | 77.3 ± 1.9 [cd] | 84.3 ± 1.9 [c] |
| α-Pinene | 5.0 | 8.3 ± 1.1 [k] | 14.3 ± 1.1 [h] | 19.4 ± 1.3 [i] | 25.4 ± 2.1 [k] |
| | 10.0 | 13.1 ± 1.1 [jk] | 20.8 ± 1.3 [gh] | 33.6 ± 1.3 [h] | 42.6 ± 1.8 [hi] |
| | 20.0 | 19.6 ± 1.6 [ghi] | 28.5 ± 2.3 [f] | 51.3 ± 1.9 [f] | 56.3 ± 1.7 [g] |
| | 40.0 | 23.0 ± 2.3 [fgh] | 44.0 ± 2.9 [e] | 70.5 ± 2.3 [d] | 77.5 ± 2.0 [e] |
| * *F*-value | - | 167.90 | 278.00 | 323.96 | 436.45 |

Each result is the mean of 6 replicates, each made with 20 adults (*n* = 120). Means within a column followed by same letters are not significantly different ($p \leq 0.05$) (Tukey's HSD test). * All *F*-values are significant, at $p \leq 0.001$.

### 3.4. Fumigation Bioactivity

Results of fumigation bioactivity (Table 4) tests demonstrated that the nanoemulsion and the crude oil exhibited the strongest fumigant bioactivity (10 µL/L air of both botanicals caused 100% adult mortality after 4 days). At these conditions, (%) mortality was 68.6, 53.3, and 51.6% for α-cedrol, α-pinene, and δ-3-carene, respectively. After 4 days of exposing weevils to 20 µL/L air, α-cedrol caused 100% adult mortality. After 7 days of exposing insects to 20 µL/L air of botanicals, the mortality of adult weevils ranged between 81.5 and 100%.

**Table 4.** Fumigant insecticidal bioactivity of *S. oryzae* exposed to essential oil, nanoemulsion, and terpene fractions of *C. sempervirens*.

| Test Material | Concentration (µL/L Air) | Mortality (% Mean ± S.E.) after Exposure Period | | | |
|---|---|---|---|---|---|
| | | **Day 1** | **Day 2** | **Day 4** | **Day 7** |
| Crude oil | 2.5 | 15.6 ± 1.3 [ijk] | 23.1 ± 1.6 [fg] | 39.3 ± 2.4 [e] | 53.8 ± 2.1 [h] |
| | 5.0 | 24.1 ± 1.3 [hi] | 36.0 ± 2.3 [e] | 63.6 ± 3.2 [c] | 72.7 ± 2.4 [e] |
| | 10.0 | 33.6 ± 2.1 [efg] | 55.3 ± 3.6 [cd] | 100.0 ± 0.0 [a] | 100.0 ± 0.0 [a] |
| | 20.0 | 55.3 ± 2.3 [c] | 94.6 ± 2.2 [a] | 100.0 ± 0.0 [a] | 100.0 ± 0.0 [a] |

**Table 4.** *Cont.*

| Test Material | Concentration (μL/L Air) | Mortality (% Mean ±S.E.) after Exposure Period | | | |
|---|---|---|---|---|---|
| | | **Day1** | **Day 2** | **Day 4** | **Day 7** |
| Nanoemulsion | 2.5 | 27.5 ± 1.1 [fgh] | 36.1 ± 2.3 [e] | 66.6 ± 2.1 [c] | 74.1 ± 2.4 [e] |
| | 5.0 | 49.9 ± 2.3 [cd] | 60.3 ± 3.6 [c] | 92.1 ± 2.1 [a] | 100.0 ± 0.0 [a] |
| | 10.0 | 68.4 ± 3.1 [b] | 94.3 ± 3.0 [a] | 100.0 ± 0.0 [a] | 100.0 ± 0.0 [a] |
| | 20.0 | 91.1 ± 3.3 [a] | 100.0 ± 0.0 [a] | 100.0 ± 0.0 [a] | 100.0 ± 0.0 [a] |
| α-Cedrol | 2.5 | 13.3 ± 1.6 [jk] | 20.6 ± 3.3 [fg] | 34.3 ± 4.5 [ef] | 49.5 ± 3.4 [i] |
| | 5.0 | 20.3 ± 2.1 [hij] | 32.3 ± 3.3 [e] | 49.3 ± 4.5 [d] | 58.9 ± 2.7 [g] |
| | 10.0 | 34.8 ± 2.1 [ef] | 47.9 ± 3.6 [d] | 68.6 ± 3.3 [c] | 76.1 ± 2.3 [e] |
| | 20.0 | 46.4 ± 2.3 [d] | 70.3 ± 4.1 [b] | 100.0 ± 0.0 [a] | 100.0 ± 0.0 [a] |
| δ-3-Carene | 2.5 | 9.3 ± 1.3 [k] | 15.3 ± 3.3 [fg] | 30.3 ± 4.5 [fg] | 40.8 ± 3.4 [j] |
| | 5.0 | 14.3 ± 2.3 [jk] | 23.0 ± 3.3 [f] | 40.0 ± 4.5 [e] | 52.5 ± 2.8 [h] |
| | 10.0 | 25.0 ± 2.0 [gh] | 37.3 ± 3.6 [e] | 53.3 ± 3.3 [d] | 64.0 ± 3.2 [f] |
| | 20.0 | 37.3 ± 2.9 [e] | 56.3 ± 4.1 [c] | 77.0 ± 3.9 [b] | 89.9 ± 2.1 [c] |
| α-Pinene | 2.5 | 8.9 ± 1.1 [k] | 12.9 ± 3.3 [g] | 26.0 ± 3.2 [g] | 33.0 ± 3.3 [g] |
| | 5.0 | 12.1 ± 1.1 [jk] | 19.1 ± 1.8 [fg] | 36.3 ± 4.5 [ef] | 47.4 ± 3.6 [ef] |
| | 10.0 | 19.6 ± 2.1 [hij] | 30.9 ± 1.5 [e] | 51.6 ± 3.1 [d] | 59.7 ± 3.1 [g] |
| | 20.0 | 33.1 ± 2.3 [efg] | 53.3 ± 2.3 [cd] | 70.1 ± 2.9 [c] | 81.2 ± 2.6 [d] |
| * *F*-value | - | 134.04 | 265.03 | 317.22 | 302.08 |

Each result is the mean of 6 replicates, each made with 20 adults ($n = 120$). Means within a column followed by same letters are not significantly different ($p \leq 0.05$) (Tukey's HSD test). * All *F*-values are significant, at $p \leq 0.00$.

### 3.5. The Dose-Response Mortality

The $LC_{50}$ and $LC_{90}$ and their confidence limits are illustrated in (Table 5). For the contact bioassay, $LC_{50}$ values of the botanicals after 48 h of treatment were: Nanoemulsion ($LC_{50} = 5.8$ μL/cm$^2$, $\chi^2 = 0.94$, $df = 4$), crude oil ($LC_{50} = 13.3$ μL/cm$^2$, $\chi^2 = 1.33$, $df = 4$), α-cedrol ($LC_{50} = 15.1$ μL/cm$^2$, $\chi^2 = 2.04$, $df = 4$), δ-3-carene ($LC_{50} = 30.7$ μL/cm$^2$, $\chi^2 = 2.77$, $df = 4$), and α-pinene ($LC_{50} = 53.4$ μL/cm$^2$, $\chi^2 = 3.12$, $df = 4$). The $LC_{50's}$ of the phytochemicals after 48 h fumigation were as follows: Nanoemulsion ($LC_{50} = 4.1$ μL/L air, $\chi^2 = 0.91$, $df = 54$), crude oil ($LC_{50} = 8.7$ μL/L air, $\chi^2 = 1.08$, $df = 4$), α-cedrol ($LC_{50} = 12.2$ μL/L air, $\chi^2 = 2.16$, $df = 4$), δ-3-carene ($LC_{50} = 17.2$ μL/L air, $\chi^2 = 3.06$, $df = 54$), and α-pinene ($LC_{50} = 19.6$ μL/L air, $\chi^2 = 3.22$, $df = 4$).

**Table 5.** * $LC_{50}$ and $LC_{95}$ and their fiducial limits of EO materials against *S. oryzae* 48 h post treatment.

| Test Material | Bioassay | $LC_{50}$ ** (95% fl) | $LC_{95}$ ** (95% fl) | Slope (±S.E.) | *** $\chi^2$ ($df = 4$) |
|---|---|---|---|---|---|
| Crude oil | Contact (μL/cm$^2$) | 13.3 (11.1–16.3) | 25.9 (19.3–32.2) | 2.1 ± 0.20 | 1.33 |
| | Fumigation (μL/L) | 8.7 (7.5–10.1) | 16.3 (13.8–21.3) | 2.0 ± 0.24 | 1.08 |
| Nanoemulsion | Contact (μL/cm$^2$) | 5.8 (5.3–7.2) | 10.2 (8.6–13.3) | 1.5 ± 0.18 | 0.94 |
| | Fumigation (μL/L) | 4.1 (3.7–4.9) | 7.3 (6.1–8.6) | 1.6 ± 0.14 | 0.91 |
| α-Cedrol | Contact (μL/cm$^2$) | 15.1 (13.4–18.9) | 27.5 (22.9–35.3) | 2.1 ± 0.28 | 2.04 |
| | Fumigation (μL/L) | 12.2 (10.5–15.8) | 22.9 (18.6–27.1) | 2.6 ± 0.26 | 2.18 |
| δ-3-Carene | Contact (μL/cm$^2$) | 30.7 (27.6–36.6) | 55.3 (48.4–64.8) | 2.9 ± 0.32 | 2.77 |
| | Fumigation (μL/L) | 17.2 (15.4–21.3) | 39.6 (34.7–47.6) | 2.8 ± 0.40 | 3.06 |
| α-Pinene | Contact (μL/cm$^2$) | 53.4 (46.3–63.1) | 114.8 (101.8–119.1) | 3.1 ± 0.30 | 3.12 |
| | Fumigation (μL/L) | 19.6 (17.3–24.5) | 42.2 (37.0–50.3) | 2.7 ± 0.41 | 3.22 |

Each result is the mean of 6 replicates, each including 20 individuals ($n = 120$). * $LC_{50}$ and $LC_{95}$ are considered significantly different when the 95% fiducial limits (f.l.) fail to overlap. ** fl = fiducial limits. *** *Chi*-square value, significant at $p \leq 0.05$ level; $df$ = degree of freedom.

### 3.6. Repellence Bioactivity

As illustrated in Table 6 and Figure 4, the EO materials strongly repelled the adult weevils, and the repellent bioactivity was both time- and dose-dependent. The crude oil of *C. sempervirens* was the strongest insect repellent, even at low concentrations, followed by

nanoemulsion, α-cedrol, and δ-3-carene; by contrast, the monoterpene α-pinene showed a weak-to-moderate repelling efficacy. The crude oil completely repelled the adult weevils at 0.44 μL/cm² after 12 h. The crude oil, nanoemulsion, and α-cedrol caused 100% repellency when the weevils were treated with a concentration equaling 0.88 μL/cm² of these products after 24 h. At this concentration, the remaining monoterpenes caused moderate repelling activities. At the lowest concentration tested (0.22 μL/cm²), the percentage repellency was 73.9, 61.3, 43.9, 33.3, and 22.1% for the EO, the nanoemulsion, α-cedrol, δ-3-carene, and α-pinene, respectively.

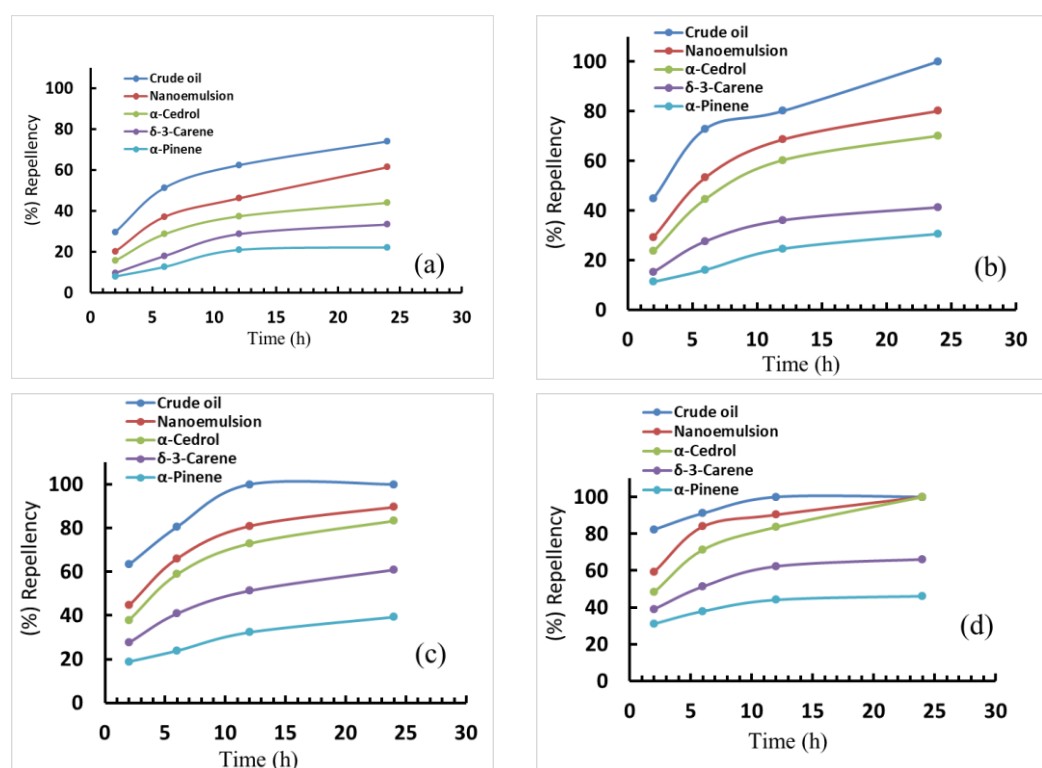

**Figure 4.** Repellent activity of cypress EO products against *S. oryzae* at (**a**) 0.11 μL/cm²; (**b**) 0.22 μL/cm²; (**c**) 0.44 μL/cm²; and (**d**) 0.88 μL/cm². (%) (Repellency in control was nil).

**Table 6.** Repellence activity of EO, nanoemulsion, and terpenes of *C. sempervirens* against *S. oryzae* adult weevils.

| Test Material | Concentration (μL/cm²) | Repellency (% Mean ± S.E.) after Period (h) | | | |
|---|---|---|---|---|---|
| | | 2 | 6 | 12 | 24 |
| Crude oil | 0.11 | 29.6 ± 1.1 [gh] | 51.3 ± 2.3 [efg] | 62.3 ± 3.1 [f] | 73.9 ± 2.6 [e] |
| | 0.22 | 44.9 ± 2.3 [e] | 72.9 ± 2.3 [bcd] | 80.1 ± 2.9 [c] | 100.0 ± 0.0 [a] |
| | 0.44 | 63.3 ± 3.1 [b] | 80.6 ± 1.9 [abc] | 100.0 ± 0.0 [a] | 100.0 ± 0.0 [a] |
| | 0.88 | 82.3 ± 3.1 [a] | 91.1 ± 2.6 [a] | 100.0 ± 0.0 [a] | 100.0 ± 0.0 [a] |
| Nanoemulsion | 0.11 | 20.1 ± 1.3 [j] | 37.1 ± 1.3 [gh] | 46.1 ± 2.3 [h] | 61.3 ± 2.1 [h] |
| | 0.22 | 29.3 ± 1.6 [gh] | 53.3 ± 2.1 [ef] | 68.6 ± 2.6 [e] | 80.1 ± 2.2 [d] |
| | 0.44 | 44.6 ± 2.1 [e] | 66.1 ± 3.6 [cde] | 80.9 ± 3.3 [c] | 89.6± 0.0 [b] |
| | 0.88 | 59.3 ± 2.6 [c] | 83.9 ± 2.6 [ab] | 90.3 ± 3.1 [b] | 100.0 ± 0.0 [a] |
| α-Cedrol | 0.11 | 15.6 ± 1.3 [k] | 28.6 ± 2.1 [hi] | 37.3 ± 2.3 [i] | 43.9 ± 2.3 [i] |
| | 0.22 | 23.6 ± 2.1 [i] | 44.6 ± 2.3 [fg] | 60.3 ± 2.1 [f] | 70.1 ± 2.1 [f] |
| | 0.44 | 37.9 ± 2.1 [f] | 58.9 ± 2.6 [def] | 72.9 ± 1.9 [d] | 83.3 ± 1.6 [c] |
| | 0.88 | 48.3 ± 2.3 [d] | 71.3 ± 3.1 [bcd] | 83.6 ± 2.1 [c] | 100.0 ± 0.0 [a] |

**Table 6.** *Cont.*

| Test Material | Concentration (μL/cm²) | Repellency (% Mean ± S.E.) after Period (h) | | | |
|---|---|---|---|---|---|
| | | **2** | **6** | **12** | **24** |
| *δ*-3-Carene | 0.11 | 9.6 ± 1.3 [lm] | 17.9 ± 1.9 [ij] | 28.6 ± 3.1 [k] | 33.3 ± 3.1 [k] |
| | 0.22 | 15.3 ± 2.1 [k] | 27.6 ± 1.6 [hij] | 36.1 ± 2.1 [i] | 41.3 ± 2.6 [j] |
| | 0.44 | 27.6 ± 2.1 [h] | 43.9 ± 2.6 [fg] | 51.3 ± 2.6 [g] | 60.9 ± 2.3 [h] |
| | 0.88 | 39.1 ± 2.3 [f] | 51.3 ± 3.3 [efg] | 62.3 ± 1.9 [f] | 66.1 ± 1.9 [g] |
| *α*-Pinene | 0.11 | 7.9 ± 1.1 [m] | 12.6 ± 1.1 [k] | 20.9 ± 1.6 [m] | 22.1 ± 2.3 [m] |
| | 0.22 | 11.3 ± 1.3 [l] | 16.1 ± 1.3 [ij] | 24.6 ± 1.9 [l] | 30.6 ± 1.9 [l] |
| | 0.44 | 18.9 ± 1.6 [j] | 23.9 ± 2.3 [hij] | 32.3 ± 1.9 [j] | 39.3 ± 2.6 [j] |
| | 0.88 | 31.1 ± 2.1 [g] | 37.9 ± 2.9 [gh] | 44.1 ± 2.1 [h] | 46.1 ± 2.3 [i] |
| Control | - | 0.0 ± 0.0 | 0.0 ± 0.0 | 0.0 ± 0.0 | 0.0 ± 0.0 |
| * *F*-value | - | 961.75 | 61.65 | 1237.03 | 2644.87 |

Each result is the mean of 5 repeats, each including 20 individuals (*n* = 100). Means within a column followed by same letter(s) are not significantly different. ($p \leq 0.05$) (Tukey's HSD test). * All *F*-values are significant at $p \leq 0.001$.

*3.7. AChE Inhibition*

All test EO materials caused a remarkable inhibition of AChE activity in *S. oryzae* (Table 7). The nanoemulsion ($IC_{50}$ = 9.88 mM, $\chi^2$ = 1.68, *df* = 5, *p* = 0.201) was the superior AChE inhibitor, followed by EO ($IC_{50}$ = 14.03 mM, $\chi^2$ = 2.41, *df* = 5, *p* = 0.243), and *α*-cedrol ($IC_{50}$ = 17.21 mM, $\chi^2$ = 2.88, *df* = 5, *p* = 0.311). By contrast, *δ*-3-carene and *α*-pinene caused moderate effects. The $IC_{50}$ of methomyl was $2.44 \times 10^{-3}$ mM.

**Table 7.** Inhibition of acetylcholinesterase (AChE) of *S. oryzae* larvae by EO materials of *C. sempervirens*.

| Plant Material | * $IC_{50}$ (mM) | (95% Fiducial Limits) | Slope (±S.E.) | ** $\chi^2$(*df* = 5) | *p* |
|---|---|---|---|---|---|
| Crude oil | 14.03 | (12.21–16.28) | 1.44 ± 0.22 | 2.41 | 0.243 |
| Nanoemulsion | 9.88 | (7.94–11.71) | 1.09 ± 0.14 | 2.68 | 0.201 |
| *α*-Cedrol | 17.21 | (15.20–20.07) | 1.30 ± 0.19 | 2.88 | 0.311 |
| *δ*-3-Carene | 34.54 | (30.00–39.33) | 1.66 ± 0.25 | 3.82 | 0.355 |
| *α*-Pinene | 39.83 | (34.44–46.12) | 2.08 ± 0.34 | 4.05 | 0.512 |
| Methomyl | $2.17 \times 10^{-3}$ | ($1.73 \times 10^{-3}$–$3.66 \times 10^{-3}$) | 1.03 ± 0.18 | 2.62 | 0.377 |

* The concentration causing 50% enzyme inhibition. ** *Chi*-square value, not significant at $p \leq 0.05$ level; *df* = degree of freedom.

*3.8. Phytotoxicity Assessment*

Phytotoxicity testing revealed that the botanicals were not phytotoxic to wheat plants, where the agronomical parameters of wheat (%) germination, and the growth of radicals and shoots were unaffected after treatment with botanicals at concentrations ranging between 50.0 and 150 μL/mL (Table 8 and Figure 5). Percentage germination and growth of shoots are slightly affected at 150 μL/mL, especially with EO, nanoemulsion, and *α*-cedrol. By contrast, the remaining compounds were nonphytotoxic, even at high test concentrations.

**Table 8.** * Phytotoxic activities of essential oil, nanoemulsion, and major fractions of *C. sempervirens* against wheat plants.

| Plant Material | Concentration (μL/mL) | Germination (%) | RL | SL |
|---|---|---|---|---|
| Crude oil | 50 | 90.6 ± 1.4 [a] | 9.08 ± 0.31 [a] | 3.32 ± 0.14 [ab] |
| | 100 | 80.3 ± 1.5 [ab] | 8.89 ± 0.23 [a] | 3.12 ± 0.11 [ab] |
| | 150 | 70.1 ± 1.3 [b] | 8.07 ± 0.18 [a] | 2.26 ± 0.15 [c] |

**Table 8.** *Cont.*

| Plant Material | Concentration (μL/mL) | Germination (%) | RL | SL |
|---|---|---|---|---|
| Nanoemulsion | 50 | 88.9 ± 1.3 [a] | 9.03 ± 0.19 [a] | 3.16 ± 0.12 [ab] |
| | 100 | 76.3 ± 1.2 [ab] | 8.70 ± 0.20 [a] | 3.03 ± 0.14 [ab] |
| | 150 | 65.2 ± 1.2 [bc] | 7.08 ± 0.28 [a] | 2.05 ± 0.12 [c] |
| *α*-Cedrol | 50 | 88.2 ± 1.3 [a] | 9.11 ± 0.41 [a] | 3.28 ± 0.15 [ab] |
| | 100 | 83.4 ± 1.9 [ab] | 9.02 ± 0.26 [a] | 3.20 ± 0.13 [ab] |
| | 150 | 74.2 ± 1.4 [b] | 8.24 ± 0.12 [a] | 2.59 ± 0.17 [bc] |
| *δ*-3-Carene | 50 | 90.6 ± 1.7 [a] | 9.20 ± 0.18 [a] | 3.39 ± 0.16 [a] |
| | 100 | 88.7 ± 1.4 [a] | 9.12 ± 0.31 [a] | 3.34 ± 0.11 [a] |
| | 150 | 88.3 ± 1.9 [a] | 9.01 ± 0.22 [a] | 3.09 ± 0.11 [ab] |
| *α*-Pinene | 50 | 91.8 ± 1.1 [a] | 9.24 ± 0.19 [a] | 3.41 ± 0.13 [a] |
| | 100 | 91.0 ± 1.3 [a] | 9.23 ± 0.08 [a] | 3.31 ± 0.11 [ab] |
| | 150 | 90.2 ± 1.3 [a] | 9.19 ± 0.22 [a] | 3.16 ± 0.12 [ab] |
| Control | - | 91.7 ± 1.6 [a] | 9.22 ± 0.32 [a] | 3.43 ± 0.18 [a] |
| *F*-value | - | 4.28 | 1.16 | 7.94 |

* Each value is the mean ± S.E. of 4 trials; RL = Radicle growth (length of seeds, cm); SL = Shoot length (cm). In a column, means followed by same letter (s) are not significantly different ($p \leq 0.05$). All *F*-values are significant at $p \leq 0.001$.

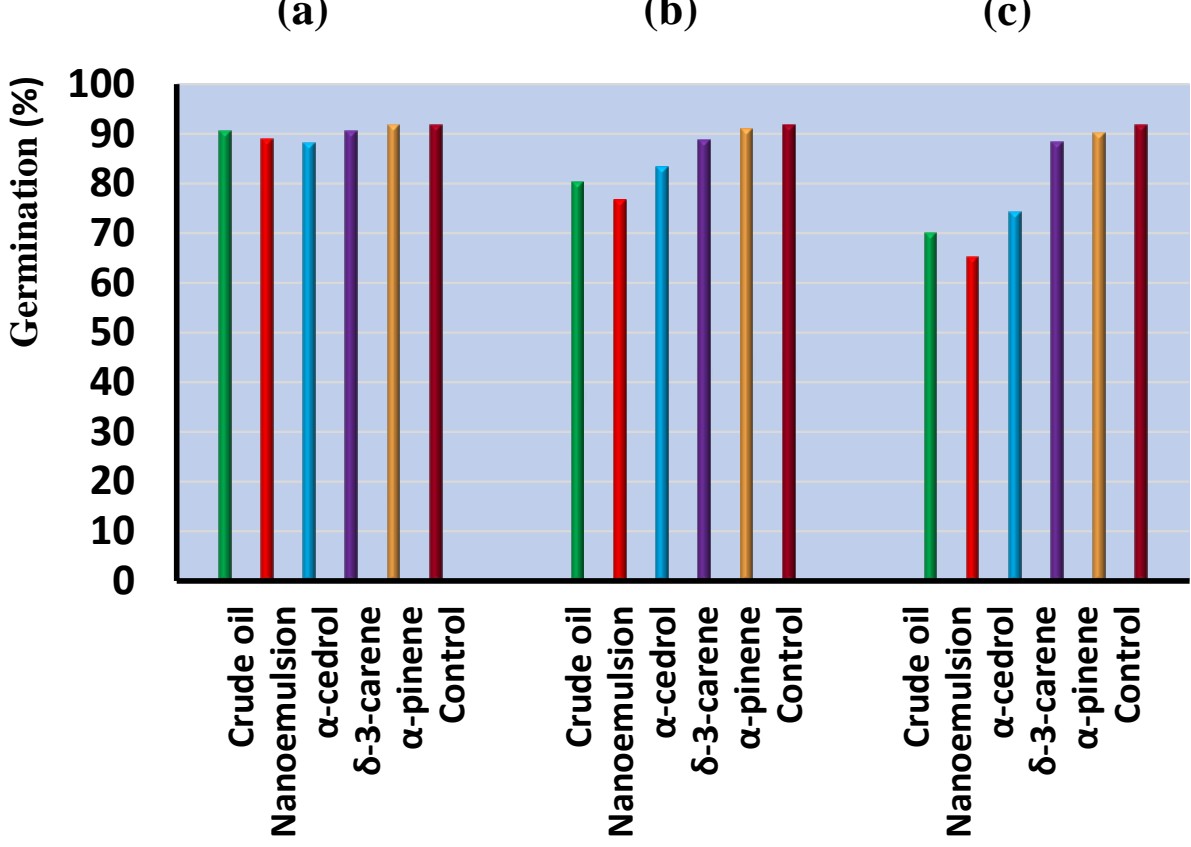

**Figure 5.** Viability of wheat grains as (%) germination treated with cypress EO products at (**a**) 50 μL/mL; (**b**) 100 μL/mL; and (**c**) 150 μL/mL.

*3.9. Toxicity against Earthworms*

The test botanicals showed relative safety toward *E. fetida*. Neither mortality nor toxicity signs were recorded in treated animals, even at 200 mg per $kg^{-1}$ soil. On the other hand, the chemical pesticide *α*-cypermethrin at 20.0 mg per $kg^{-1}$ caused 100% mortality of earthworms after 10 days.

## 4. Discussion

The yield and composition of *C. sempervirens* var. *horizontalis* oil are in a good accordance with previous studies of the same Saudi species or those of other similar flora, where *α*-pinene was the main component of cypress EO [21,39–42]. Variations both in the yield and the abundant oil terpenes of plant oils have been recorded in previous reports [22,43–45]. However, *β*-thujene was presented as the main terpene component (31.4%) in the EO of Brazilian *C. sempervirens* [43]. These variations are mainly dependent on many factors, including genetic and geographic factors. The soil status, the method of cultivation, water availability, seasonality, the extracted parts, and extraction techniques are also of major influence [6,46–48].

According to our findings, the EO of cypress, its nanoemulsion, and individual terpenes exhibited remarkable insecticidal, repellent, and AChE effects against *S. oryzae*. To our knowledge, bioactivity of the EO of cypress belonging to Saudi flora, particularly its nanoemulsion and individual terpenes, had not been investigated against insects of stored grain; hence our study is considered a first report. A remarkable fumigation bioactivity of the EO extracted from Egyptian cypress was recorded against adults of *S. oryzae* with $LC_{50}$ = 17.2 mg/L air.

The EO of *C. sempervirens* was reported to possess toxic and repellent bioactivities against *Sitophilus zeamais* and *Tribolium confusum* using the impregnated filter paper bioassay, as well as treated grains [22]; a remarkable repellent potential against the codling moth, *Cydia pomonella*; and moderate toxicity and repellent activities against the mosquitoes, *Aedes albopictus* and *Ae. aegypti* [49–51]. The oil materials tested herein strongly repelled *S. oryzae* adult weevils. There are many factors affecting the repellent bioactivity of the plant-based products against harmful insects, which depend mainly on the nature of products under investigation; the respiratory system upon which the plant bioactive substances act, and the insect's olfactory receptors are of a major influence [52]. The high volatile nature of EOs play a main role in this phenomenon, where they can be inhaled, ingested, or easily absorbed through the insect's skin [53,54]. The repellence bioactivity of cypress EO products toward *S. oryzae* was in accordance with studies and reports that investigated the bioinsecticidal and repellant potential of plant EOs against insects of stored grain, including *S. oryzae* [10,11,55–57].

In this study, a green approach was followed to prepare an oil-in-water nanoemulsion (droplet size 67.8 ± 3.1 nm) from cypress oil using fewer toxic chemicals at acceptable concentrations, in the proportions 5:90:5% (EO:$H_2O$:Tween 80 as an emulsifier). Tween surfactants, especially Tween 80 and Tween 20, are frequently utilized as emulsifiers in the preparations of oil/water nanoemulsions as they can produce stable formulations without using cosurfactants [58]; However, Tween 80 was selected herein as a nonanionic emulsifier, due to its miscibility with water and good solubility for EOs. It is characterized by a high hydrophilic–lipophilic balance (HLB = 15); hence, decreasing the tension between the oil and aqueous phases, and resulting in the formation of stable emulsions [17]. In most cases, Tween 80 also appeared to perform better than Tween 20 in terms of droplet size distribution and the stability of the nanoemulsion, which may be due to the structural differences in the nonpolar tail of the two molecules [59,60]. In both micro- and nanoemulsion preparations, the surfactant functions to reduce the interfacial energy by providing a mechanical barrier to coalescence [17]. The nanoemulsion of cypress oil exhibited a good stability up to 45 days after preparation when exposed to stress conditions during storage. Meanwhile, nonequilibrium emulsion formulations may undergo a breakdown, resulting in sedimentation, flocculation, and coalescence, resulting in many shortcomings in their biological activities. Alternatively, because of their novel properties, such as subcellular size, nanoemulsions have good stability under extreme conditions [17,61]. The pH of the nanoemulsion stabilized around 6.5 during storage. The pH of an emulsion is critical to its stability because changes in pH affect the surface charge of the globules, disrupting their stability. Furthermore, increases in the surface charge of globules cause electrostatic repulsion, which reduces flocculation and leads to the dissolution of micro- and nanoemul-

sions [17]. In a nanoemulsion formulation, the PDI determines droplet size stability and uniformity; a low PDI ensures high droplet size uniformity. Over 30 days of storage, the PDI of cypress oil nanoemulsion ranged between $0.18 \pm 0.03$ and $0.24 \pm 0.02$. Many authors have reported that a PDI of less than 0.25 indicates a narrow distribution of particle size, providing stability and homogeneity due to a reduced Ostwald ripening [17,61].

The nanoemulsion of cypress oil exhibited superior bioactivity against the target weevil. When materials are formulated at the nanoscale, they acquire novel chemical and physical properties, such as increased surface area, solubility, and high affinity to the targeted biosystems, which promotes their biological activities [17]. Because of these novel criteria, nanomaterials are promising candidates for developing effective eco-friendly insecticides. To avoid the overuse of the toxic solvents or high-energy inputs that are commonly used in pesticide synthesis, the "green synthesis" concept has been proposed, outlining the potential use of animal, microbial, and plant-borne compounds as stabilizing agents for the production of bioactive nanomaterials [15,62]. As a result, nanotechnology is being considered as an alternative strategy to improve the stability and bioactivity of pesticide materials that rely on various nanocarriers, such as plant-oil-based nanoemulsions [6,15,17,63]. In the literature, the reported insecticidal activity of plant-based nanopesticides, including oil nanoemulsions against serious insects, such as those infesting stored grains, has been reported [6,15,16,18]. Nenaah reported that nanoemulsions made from the EOs of three Achillea species, *A. biebersteinii*, *A. santolina*, and *A. millefolium*, outperformed their bulk counterparts in adulticidal activity against *T. castaneum* [6]. Similar results have been reported for *T. confusum* and *Cryptolestes ferrugineus* [18,63].

The bioactivity of plant oils are attributed to several components, with demonstrable insecticidal activity contained in the plant EO, especially in monoterpenes such as $\alpha$-pinene, $\alpha$-terpinene, limonene, camphor, carvacrol, thymol, $\delta$-3-carene, $\alpha$-thujone, 1,8-cineol (eucalyptol), eugenol, and ascaridole [3,6,9–11,17,57]. Although a synergism with other minor constituents is common where each oil component participates in penetration, fixation, and distribution into biomembranes [6,9,17], synergy between components of an EO might be occur between several components contained in the same oil, or between different essential oils with known biological activities [17,64,65].

Essential oils, particularly monoterpenes, are volatile and lipophilic, allowing them to quickly penetrate the integument of insects, interfering with physiological parameters and causing alteration in all vital functions. [6,9,17,53]. The EO materials caused a considerable inhibition in the AChE bioactivity of *S. oryzae*, indicating a neurotoxic mechanism of action. As mixtures, the toxicity of EOs is not yet fully understood. Nevertheless, the rapid action against some pests is major evidence of a neurotoxic action, which is attributed to AChE inhibition, as described herein [9–12,17]. Comparing our results with previous reports, the dichloromethane, acetone, ethyl acetate, and methanol extracts of the cones and leaves of *Cupressus sempervirens* var. *horizantalis* displayed a moderate inhibition of butyryl-cholinesterase, AChE, and tyrosinase bioactivities at 200 µg/mL [66]. Aazza reported the acetylcholinesterase inhibitory effect of cypress oil (where $IC_{50}$ was 0.2837 mg/mL) using bovine acetylcholine [67]. Recently, Alimi et al. found that the EO of cypress displayed a significant inhibition in AChE activity of *Hyalomma scupense* (Acari: Ixodidae) [68]. The plant EOs can interfere with other protein targets, which disrupts the insect's nervous system, such as the nicotinic acetylcholine receptors (nAChR), and the octopamine or the neurotransmitter inhibitor $\gamma$-aminobutyric acid (GABA). EOs were found to inhibit enzymatic biosystems (superoxide dismutase (SOD), catalase (CAT), glutathione-S-transferase (GST), and glutathione reductase (GR)), peroxidases (POx), and the nonenzymatic (glutathione (GSH)) antioxidant defense biosystems [9,11].

According to our findings, the EO materials showed a relative safety within the limit of the test concentrations when tested on *E. fetida* (a common earthworm) and wheat plants. Plant products with pesticidal activities are often wrongly considered safe with no negative effects on nontargets, including humans, without this being experimentally verified. Nevertheless, many authors stated that EOs, nanoemulsion preparations, and oil

terpenes were relatively safe when assessed against several nontarget species [3,17,28,69]. In that regard, most reports have focussed on assessing the acute toxicity, whereas both subchronic and chronic evaluations have not been fully undertaken [5,17,69–71]. Regarding cypress oil, health risks or side effects following administration of designated therapeutic dosages are not recorded. Nevertheless, kidney irritation was recorded with the intake of large doses [72]. However, not all natural products are free of risk, therefore deep investigations are always required to explore the biosafety of the plant-based pesticides before practical use in stored-product insect control programs. The authors should discuss the findings and how they can be interpreted in light of previous research and the working hypotheses. The findings and implications should be discussed in the broadest possible context. Future research directions may be highlighted as well.

## 5. Conclusions

According to the results of the present study, cypress EO, the oil nanoemulsion, and its individual terpenes showed remarkable insecticidal, repellence, and acetylcholinesterase inhibitory bioactivities against the rice weevil, *S. oryzae*. There were no significant adverse effects on the earthworms, nor the agronomical parameters of wheat plant. When properly prepared, cypress oil, its nanoemulsion, and its main terpenes could be applied as novel ecofriendly natural pest-control options against *S. oryzae*, being more appropriate than the chemical insecticides. However, deep toxicological evaluations should be carried out to substantiate the relevant concentrations and adverse effects of the test products against mammals and other nontarget organisms.

**Author Contributions:** Conceptualization, G.E.N.; methods, investigation, and validation, G.E.N., B.Z.A., A.A.A., D.M.S. and S.A.; analysis, G.E.N.; resources, curation, G.E.N., B.Z.A., S.A., D.M.S. and A.A.A.; writing original draft, G.E.N.; review and editing, G.E.N. and A.A.A.; supervision, G.E.N. All authors have read and agreed to the published version of the manuscript.

**Funding:** This research received no external funding.

**Institutional Review Board Statement:** The study was conducted in accordance with the Declaration of Helsinki, and approved by the Institutional Ethics Committee.

**Informed Consent Statement:** Not applicable.

**Data Availability Statement:** Data supporting the conclusions of this article are presented in the main manuscript.

**Conflicts of Interest:** The authors declare no conflict of interest.

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
