# Peer review of "Cupressus sempervirens Essential Oil, Nanoemulsion, and Major Terpenes as Sustainable Green Pesticides against the Rice Weevil"

_sustainability, doi:10.3390/su15108021_

Round 1
Reviewer 1 Report
The document is interesting. The document may be accepted for publication after some minor corrections.
Some suggested minor revisions:
Line 25 – “S. oryzae” in italic
Line 60 – delete a full stop in this sentence “on food products..”
Line 99 – change “for mor “ to ”for more”
Line 154 – Delete one coma “(wt./wt.),, “
Line 163 – delete one coma “turbidity, , and”
Line 169 – I think something is missing here “25 0.2°C,”
Line 201/202 – I think something is missing here “After evaporating acetone from the, each…”
Line 317 - caption the title of the tables in the same way “Table (2) Characterization …”
Line 503 – delete one full stop “and ?-cedrol..”
Line 573 – delete full stop “treated grains [22]., a”
Line 757 – Delete coma after the two points “Corporation:, Carol Stream,”
Review the references. They are not all the same.
Some examples:
Number 10 (line 722) – “J Pest Sci” – change to “J. Pest Sci.”
Number 71 (line 861) – Biologica Control – change to “Biol. Control”
Author Response
Dear Dr.,
I would like to thank you and the reviewers for all the valuable comments and constructive suggestions on the Manuscript ID: sustainability-2359826, titled "Cupressus sempervirens Essential Oil, Nano-emulsion and Major Terpenes as Sustainable Green Pesticides Against the Rice Weevil". In this revised form of the manuscript (R1), I considered all comments of the editor and reviewers. Please find each of these comments in conjugation with my response (point by point). Furthermore, all of the corrected words and/or statements are highlighted in a red color in the revised manuscript.
Please find attached the revised version of the manuscript, which I would like to submit for your kind consideration.
-------------------------------------------------------------------------------------------------------
Reviewer 1
Comment:
Line 25 – “S. oryzae” in italic
Response: (Done).
Comment:
Line 60 – delete a full stop in this sentence “on food products..”
Response: (Done).
Comment:
Line 99 – change “for mor “ to ”for more”
Response: (Done).
Comment:
Line 154 – Delete one coma “(wt./wt.),, “
Response: (Done).
Comment:
Line 163 – delete one coma “turbidity, , and”
Response: (Done).
Comment:
Line 169 – I think something is missing here “25 0.2°C,”.
Response: Thank you for this valuable comment (corrected).
Comment:
Line 201/202 – I think something is missing here “After evaporating acetone from the, each…” (Thank you)
Response: Thank you for this valuable comment (corrected).
Comment:
Line 317 - caption the title of the tables in the same way “Table (2) Characterization …”
Response: (Done).
Comment:
Response: (Done).
Comment:
Line 503 – delete one full stop “and ?-cedrol..”
Response: (Done).
Comment:
Line 573 – delete full stop “treated grains [22]., a”
Response: (Done).
Comment:
Line 757 – Delete coma after the two points “Corporation:, Carol Stream,”
Response: (Done).
Comment:
Review the references. They are not all the same.
Some examples:
Number 10 (line 722) – “J Pest Sci” – change to “J. Pest Sci.”
Number 71 (line 861) – Biologica Control – change to “Biol. Control”
Response: All references have been reviewed(Done).

Reviewer 2 Report
The manuscript is well written. However, the following points can be improved:
Line 54: Insecticidal, repelling, antifeeding have synthetical pesticides as well. Please reformulate this sentence
Line 301. The graphic is blurred, it would be great if you change the quality of the image
Author Response
Dear Dr.,
I would like to thank you and the reviewers for all the valuable comments and constructive suggestions on the Manuscript ID: sustainability-2359826, titled "Cupressus sempervirens Essential Oil, Nano-emulsion and Major Terpenes as Sustainable Green Pesticides Against the Rice Weevil". In this revised form of the manuscript (R1), I considered all comments of the editor and reviewers. Please find each of these comments in conjugation with my response (point by point). Furthermore, all of the corrected words and/or statements are highlighted in a red color in the revised manuscript.
Please find attached the revised version of the manuscript, which I would like to submit for your kind consideration.
----------------------------------------------------------------------------------------
Reviewer 2
Comment:
Line 54: Insecticidal, repelling, antifeeding have synthetical pesticides as well. Please reformulate this sentence
Response: (Done).
Comment:
Line 301. The graphic is blurred, it would be great if you change the quality of the image
Response: (Done).
Reviewer 3 Report
Very interesting study that provides evidence for the potential of using essential oils as safe and alternative for chemical fumigation based on Cupressus sempervirens var. horizontalis and nano-emulsion the storage weevil Sitophilus oryzae. This study provided different aspect of research from entomological bioassays (different mode of application) to formulation with characterization of new bioproducts. Also these bioactivities of oil, nano-emulsion, and bioactive terpenes were tested for their side effects on earthworm and phytotoxicity on plants and grains. This study showed that EO of C. sempervirens and nano-emulsion was very effective against the weevil, S. oryzae with no side effects on grain, plants and earthworm.
More statistical explanation should be indicated in your study, either in the methodology and in results section. Please provide more information on how ANOVA was employed (one, two, or multi ways). Also, please indicate if the data are converted before the statistical analysis. In your statistical analysis, if the control was take it into account? If the mortality has been adjusted, why is the control included in the tables?
Also the conclusion part should be improved.
Please find attached some concerns, questions and recommendations to be addressed.

Moderate editing of English language is needed.
Author Response
Dear Dr.,
I would like to thank you and the reviewers for all the valuable comments and constructive suggestions on the Manuscript ID: sustainability-2359826, titled "Cupressus sempervirens Essential Oil, Nano-emulsion and Major Terpenes as Sustainable Green Pesticides Against the Rice Weevil". In this revised form of the manuscript (R1), I considered all comments of the editor and reviewers. Please find each of these comments in conjugation with my response (point by point). Furthermore, all of the corrected words and/or statements are highlighted in a red color in the revised manuscript.
Please find attached the revised version of the manuscript, which I would like to submit for your kind consideration.
-------------------------------------------------------------------------------------------------------
Reviewer 3
Comment:
More statistical explanation should be indicated in your study, either in the methodology and in results section. Please provide more information on how ANOVA was employed (one, two, or multi ways). Also, please indicate if the data are converted before the statistical analysis. In your statistical analysis, if the control was take it into account? If the mortality has been adjusted, why is the control included in the tables?
Response: The authors are greatly appreciated to the reviewer for these valuable comments. All these issues are considered in the revised manuscript (Done).
Comment:
Also the conclusion part should be improved.
Response: (Done).
Comment:
Moderate editing of English language is needed.
Response: A thorough revision of the English language has been made throughout the whole manuscript (Done)
Comment:
Line 25: please give full name of the insect in the keywords.
Response: (Done).
Comment:
Line 36: please reformulate the sentence, infestation caused by what? please specify.
Response: (Done).
Comment:
Line 38: delete point after condition.
Response: (Done).
Comment:
Line 45: Please mention the resistance of Sitophilus oryzae to insecticides. Please give example of insecticides resistance for this weevil with active ingredient.
Response: (Done).
Comment:
Line 60. Delete point.
Response: (Done).
Comment:
Line 89: You indicate white albino rats. However, your study does not cover its side effect on rat. Please delete it. From line 89 and from conclusion.
Response: Please accept my apology for this typo-error (corrected).
Comment:
Line 90: please specify more about the aim of the study, impact of the tested bioproducts on wheat plant (grain or plant or both…).
Response: (Done).
Comment:
Line 98: Test insect, provide the origin of Sitophilus oryzae colony. Where did you collect the insect (from field or laboratory).
Response: Thank you (Done).
Comment:
Line 103: Add point at end of the sentence.
Response: Thank you (Done).
Comment:
Line 105: please provide more detail about aerial parts of the tree (leaves with stems or just leaves)
Response: (Done).
Comment:
Line 141- 142: please reformulate this sentence and delete colon.
Response: (Done).
Comment:
Line 169: please add plus–minus sign, ± after 25°C
Response: Thank you (corrected).
Comment:
185-186: please give the name of each botanical and reformulate the sentence. You mention 0.398, 0.795, 1.59 and 3.18 ml of each botanical? was dissolved in 5 ml acetone in order to prepare the test solutions.
Response: The sentence has been corrected as follows:
For this purpose, 0.398, 0.795, 1.59 and 3.18 ml of EO, nanoemulsion, α-cedrol, δ-3-carene, and α-pinene were dissolved in 5 ml acetone in order to prepare the test solutions.
Many thanks
Comment:
193: please clarify which food- enriched Petri dishes
Response: The sentence is re-written and clarified as follows:
After 24 h, insects were introduced to clean Petri dishes enriched with sterilized wheat grains (Done).
Comment:
Line 209: It preferable to add figure/photo that explain the whole experiment
Response: At this moment, it is difficult to add a photo describing the experiment. Please accept my apology.
Comment:
238: please reformulate the sentence, delete against. Replace with was evaluated on wheat plant
Response: (Done). Thank you
Comment:
- Please start new sentence from wheat grains
Response: (Done).
Comment:
246: change sentence with : additionally, 10 ml of water was given daily if needed.
Response: (Done).
Comment:
263: please give more details about ANOVA used (one or two or muti ways). Please indicate if the data is transformed before the statistical analysis.
Response: (Done).
Comment:
Did you include the control in your statistical analysis? Why is the control included in the tables if the mortality has been corrected?
Response: Data of mortality were adjusted for control mortality and corrected using Abbott's formula when mortality in control exceeded (5%). I am sorry for this typo-error.
Comment:
Line 242: please correct with days from 2 to 7.
Response: (Done).
Comment:
Line 501-502: please reformulate the sentence, do not start with (%)
Response: (Done).
Comment:
Line 505: please change against with on the agronomical parameters of wheat plants.
Response: (Done). Thank you
Comment:
Line 533: Please replace this figure with one that is clearer. Treatments are not writing very clear.
Response: (Done).
Comment:
Line 539: table with too many 0.00 is not very attracting. If possible, to change table with figure.
Response: It is not possible to replace this table with a figure because there was no mortality in the earthworms along the testing period. However, the table can be deleted and the data may be mentioned only as a text.
Comment:
Line 559: change for example with, however. Please reformulate this sentence.
Response: (Done).
Comment:
Line 566 to 569: delete this sentence. It just repetition.
Response: All the paragraph is reformulated.(Done).
Comment:
Line 273: Please change the sentence by starting with: A remarkable.
Response: (Done).
Comment:
Line 286: delete without using toxic chemical (Tween used is toxic). It can be less toxic at acceptable concentration.
Response: (Done). Thank you
Comment:
Line 650: please give more detail about Aazza findings which insect….
Response: More details are added (Done).
Comment:
Line 662-664: please delete the whole sentence. No place for it in this context. To let you know, evaluating your bioproducts against E. fetida earthworm or on the albino rat don’t make it 100% safe. In the context of IPM, a product to be safe need to be evaluated against a large number of natural enemies’ species from different orders, with a lot of testing for its toxicology… However, it seems very safe from your findings and very promising. But not conclusive.
Response: The sentence is re-written as follows:
According to our findings, the EO products showed relative safety when assessed against the earthworm, E. fetida, and the wheat plant within the limit of the tested concentrations.
Comment:
Line 669-673: please delete this sentence. Irritation and pollen are not the aim of this study.
Response: (Done).
Comment:
Line 678-681: please delete this sentence.
Response: (Done).
Comment:
Line 685: please delete albino rats. There is no rat study according to your findings. Only earthworms were studied.
Response: Deleted. Thank you (Done).
Comment:
Line 688-689: please delete this sentence. Conclusion part should be improved.
Response: The conclusion part is re-written and improved (Done).
Reviewer 4 Report
Discussion - the most significant results should be discussed.
Conclusions - too general. Authors should focus on the most significant data.
There are some technical errors in the manuscript.
Author Response
Dear Dr.,
I would like to thank you and the reviewers for all the valuable comments and constructive suggestions on the Manuscript ID: sustainability-2359826, titled "Cupressus sempervirens Essential Oil, Nano-emulsion and Major Terpenes as Sustainable Green Pesticides Against the Rice Weevil". In this revised form of the manuscript (R1), I considered all comments of the editor and reviewers. Please find each of these comments in conjugation with my response (point by point). Furthermore, all of the corrected words and/or statements are highlighted in a red color in the revised manuscript.
Please find attached the revised version of the manuscript, which I would like to submit for your kind consideration.
-------------------------------------------------------------------------------------------------------
Reviewer 4
Comment:
Discussion - the most significant results should be discussed.
Response: (Done).
Comment:
Conclusions - too general. Authors should focus on the most significant data.
Response: (Done).
Comment:
There are some technical errors in the manuscript.
Response A thorough revision both scientifically and linguistically has been made (Done)